# Identifying and Supporting Students with a Chronically Ill Family Member: A Mixed-Methods Study on the Perceived Competences and Role Views of Lecturers

**DOI:** 10.3390/ijerph20064978

**Published:** 2023-03-11

**Authors:** Hinke M. van der Werf, Wolter Paans, Anneke L. Francke, Petrie F. Roodbol, Marie Louise A. Luttik

**Affiliations:** 1Research Group Nursing Diagnostics, Hanze University of Applied Sciences, Petrus Driessenstraat 3, 9714 CA Groningen, The Netherlands; 2Department of Critical Care, University Medical Centre Groningen, 9714 CA Groningen, The Netherlands; 3Nivel, Netherlands Institute for Health Services Research, 3513 CR Utrecht, The Netherlands; 4APH Amsterdam Public Health Research Institute, Vrije Universiteit Amsterdam, 1081 HV Amsterdam, The Netherlands; 5Faculty of Medical Sciences, University Medical Center Groningen, 9714 CA Groningen, The Netherlands; 6Research Group Family Care, Hanze University of Applied Sciences, 9714 CA Groningen, The Netherlands

**Keywords:** young carers, students, support, identifying, lecturers, higher education

## Abstract

Young adult caregivers experience reduced wellbeing when the combination of family care and an educational program becomes too demanding. We aim to clarify the role views, competences, and needs of lecturers regarding the identification and support of these students to prevent negative mental health consequences. A mixed-methods explanatory sequential design was used. We collected quantitative data using a survey of lecturers teaching in bachelor education programs in the Netherlands (*n* = 208) and then conducted in-depth interviews (*n* = 13). Descriptive statistics and deductive thematic analyses were performed. Most participants (70.2%) thought that supporting young adult caregivers was the responsibility of the educational institution, and 49% agreed that it was a responsibility of the lecturer, but only 66.8% indicated that they feel competent to do so. However, 45.2% indicated that they needed more training and expertise to identify and support these students. All interviewees felt responsible for their students’ wellbeing but highlighted a lack of clarity regarding their role fulfillment. In practice, their ability to identify and support these students depended on their available time and level of expertise. The lecturers required agreements on responsibility and procedures for further referral, as well as information on support and referral opportunities, communication skills courses, and peer-to-peer coaching.

## 1. Background

Mental health issues are common among students aged 18–25 years [1,2,3]. Results from the World Mental Health Survey conducted by the World Health Organization indicate that around 20% of students have at least one mental health disorder, as defined by the *DSM-IV* [1]. Mental health problems and reduced psychological health in students is attributed to academic pressure and the adult-like responsibilities of becoming a professional [4]. Coping with this pressure and these responsibilities can be especially stressful due to the life stage of the students. Students aged 18–25 years are expected to develop their own identities, though they might not yet have the skills and cognitive capacities of an adult [4,5]. Most mental health disorders begin during this life stage [6].

A student with a chronically ill parent (or other close family member) potentially faces additional disadvantages. Their family situation might limit the emotional or financial support that they receive from their parents or family [7]. The limited family support, worries about their family situation, specific concerns related to their own stage of life, and the associated academic pressures all combine to make these students especially vulnerable to mental health problems [8,9,10].

Mental health problems may lead to procrastination, non-attendance in lessons, delays in completion of studies, or even resignation from courses [11,12,13]. Several studies have described lecturers as ideally suited for early identification and support of these students and best-placed to prevent the abovementioned consequences of mental health problems [5,8].

However, studies of students with chronically ill family members report that lecturers often do not recognize these problems among their students or indeed understand them as related to caring for a chronically ill family member. For this reason, these students often receive no appropriate support from their schools [14,15,16]. A recent study on teachers’ perspectives on the provision of support for young carers (age < 18 y) shows the difficulties teachers experiencing while identifying young carers and balancing the provision of extra support within the constraints of the school context [17]. This lack of recognition and support may be due to a lack of knowledge and competence, as has been reported in studies of lecturers [18,19,20]. A quantitative study by Gulliver et al. [21] addressed the importance of lecturers’ literacy on mental health problems among students. They found that university staff who were knowledgeable about depression were more likely to provide support to students with mental health issues. The authors call for additional research into the attitudes of lecturers regarding the identification of mental health issues and the actions of lecturers when confronted with affected students.

The aim of this study is to gain insights into the role views, perceived competences, and needs of lecturers at universities of applied sciences in the Netherlands regarding the identification and support of students with chronically ill family members. We define a “role view” as “a settled way of thinking about the role of lecturers in terms of identifying and supporting students with chronically ill family members”. In this study, it is assumed that identification and support is relevant for both the students who are already experiencing (mental) health issues and those who are not yet affected.

## 2. Method

A mixed-methods explanatory sequential design was used [22]. Quantitative data were collected via a survey of lecturers, and qualitative data were collected through in-depth interviews with selected lecturers to obtain a more detailed explanation or underpinning of the quantitative results (Figure 1).

### 2.1. Survey: Study Sample and Recruitment

The study sample consisted of lecturers working at one of the three selected universities of applied sciences offering bachelor education programs in the Netherlands. Information about the study and a request for lecturers to participate was posted on the intranet of the universities in question. These intranet pages generally contain education-related information for the employees of these organizations. Two weeks after the first post, a reminder was posted. The intranet is frequently visited by employees, and it therefore seemed an appropriate medium to reach potential participants.

This approach was expedient, compared to selecting a targeted sample, as it allowed virtually all lecturers to participate. There were no specific inclusion criteria, except that the lecturers needed to have teaching responsibilities. The study was described as focusing on the identification and support of students with chronically ill family members. The lecturers were invited to indicate a willingness to participate by clicking on a web link included in the message. Completing the survey took approximately 10 min.

The data were collected between September and November 2020. The participants’ characteristics are summarized in Table 1.

### 2.2. Survey: Data Collection and the Content of the Survey Questionnaire

The online survey consisted of questions concerning the lecturers’ demographic characteristics and the study aim. The questions sought to elicit the participants’ views on their personal roles as lecturers and the role of universities and schools in identifying and supporting students with chronically ill family members. Finally, the lecturers were asked about their own level of competency regarding the identification and support of these students.

The survey questions were based on relevant literature regarding the role views and perceived competences of lecturers and derived from topic list of Reinke et al. [23] and Gulliver et al. [21], who investigated teachers’ perceptions about their students’ mental health. Face validity was established in the first draft of the survey with the support of experts in the field of education (*n* = 4) and family care (*n* = 4). Prior to data collection, a draft survey was pilot-tested among 10 lecturers for comprehensibility and feasibility. The content was discussed and adjusted in response to their feedback.

### 2.3. Survey: Data Analysis

The quantitative data from the survey were analyzed using the Statistical Package for the Social Sciences (SPSS), version 25. Descriptive statistics were used to describe the study population and the attitudes of lecturers.

### 2.4. Interviews: Study Sample and Recruitment

In the final question of the survey, the lecturers were asked whether they would be willing to participate in a follow-up interview. Those who agreed to participate (*n* = 52) were asked to share their email address. Subsequently, a purposeful sample of 13 lecturers was selected for the interviews, representing a range of educational programs (non-health-related and health-related fields of education), age groups, and years of experience in education, as well as both genders. The descriptive characteristics of the interview participants are summarized in Table 2.

### 2.5. Interviews: Data Collection and Interview Guide

A semi-structured interview guide was created based on the survey results, covering the main topics—namely, role views on identifying and supporting students with chronically ill family members, as well as the respondents’ perceived level of competence in doing so. Face validity was established for the first draft of the interview guide with the support of experts in the field of education (*n* = 4) and family care (*n* = 3). Two test interviews were conducted to judge the relevance of the topics and to improve the interview guide.

### 2.6. Interviews: Data Analysis

The interviews were audio-recorded, transcribed verbatim, and anonymized prior to analysis. Deductive thematic analysis was performed, guided by the topics of the interview guide and practically supported by the use of the software program ATLAS.ti 9.0.15. We followed the six steps of thematic analysis described by Braun and Clarke [24]. During the first step of the data analysis, the transcripts were read for the purposes of familiarization (1). Second, sections of the texts related to the main topics were identified. These text sections were indicated using codes, identified by HMW and MLAL (2). Overlapping codes were further refined and grouped into themes by researchers HMW, MLAL, and WP (3) and then individually and critically examined (4). These themes were discussed within the author group for accurate naming (5). Finally, the themes were reported in the results section (6).

### 2.7. Ethical Considerations

The Hanze Ethical Advice Commission—Ethical Review Board (number heac.2020.004) approved the study. Online informed consent was obtained prior to participation after participants had been informed about the aim and procedures of the study. The participants were aware that participation was voluntary and that they could withdraw at any time—or choose not to answer certain questions—without providing a reason.

## 3. Results of Survey

A total of 208 lecturers completed the questionnaire and were thus included in the study. All participants were involved in teaching students. Most (65%) indicated that they had experience of teaching students with chronically ill family members. Most (67.3%) were working in nursing or social work educational programs. A skewed distribution was found for gender and age, with most participants being female (76%) and older than 40 (61.1%). Furthermore 13 lecturers were included for interviews. Nine of the 13 interview participants were lecturers working in a health-related educational programs such as social work and nursing (Table 2). All participants had experience with identifying and supporting students with chronically ill family members and could state whether this support had been adequate.

### 3.1. Role Views and Role Fulfillment 

Questions regarding “role views” were concerned with the role of the lecturer in identifying students with chronically ill family members. Most survey respondents (86%, *n* = 179) agreed with the statement that they usually asked about a student’s family situation when the student indicated reduced wellbeing. Just over half of the lecturers (53.8%, *n* = 112) who usually asked about students’ family situations were working in health-related educational programs. The respondents were also questioned about how they would identify a student with a family member suffering from a serious chronic illness. Only 8% (*n* = 17) of the respondents indicated that students themselves would initiate a conversation, and 29.8% (*n* = 62) said that they would initiate a conversation with a student whom they were worried about. However, most (59.1%, *n* = 123) reported experiences of both situations, with either themselves or a student initiating a conversation.

Like most of the survey respondents, the interview participants indicated that they were unclear about their roles in identifying and supporting students with chronically ill family members. The range of required tasks was not clearly stated in their role descriptions, which raised questions about who should bear responsibility for these students. In practice, their individual competences determined whether they engaged in this role and how.

Like 29.9% of the survey respondents, the interview participants said that they usually asked about a student’s family situation when the student indicated reduced psychological health. They stated that identifying threats to the wellbeing of their students was part of their role as lecturers and important for preventing mental health problems and ensuring that the students remained in education:


*“I think a willingness to identify these issues and explore them in more depth should be part of your responsibilities as a lecturer. You can then figure out whether you can do something to help or refer the student to someone else who can.”*
(p12)

The participants described active and passive ways of identifying students with family members living with chronic illness. All of the interviewees described active identification as requiring an attentive attitude of the lecturer, who should be observing the students and be aware of early signs, such as absences from class, reductions in school performance, and decreased involvement in the group:


*“I notice they tend to step back and fade into the background a bit more. They’re just kind of absent. They respond more emotionally when you ask them to do something. You get kind of a ‘don’t ask too much of me, I just can’t cope right now’ vibe.”*
(p8)

Another form of active identification was described as directly asking students about their family situations in the context of an introductory or coaching meeting, for example:


*“So when you meet them first, you obviously ask, ‘How are you doing? Is the degree program still a good fit for you?’ But I do try to feel them out a bit, figure out if they’re comfortable in their own skin. Sometimes I ask them if there’s anything going on that’s been demanding more of their energy, if they’re still living at home or not, things like that. Just some general questions to get a feel for how they’re doing, which can be tricky. There’s usually some underlying issue, so you try to find out more. If students make an offhand comment about their home life, I might put them on the spot a bit and just ask them point blank what’s going on.”*
(p2)

The participants referred to passive identification as occurring when the students themselves initiated a conversation or implicitly reported about their family situation:


*“She mentioned it herself, so I just followed up on it.”*
(p1)

In line with the majority of the survey respondents, the interview participants stated that few students with chronically ill family members initiated such conversations. Initiation seemed to be more common with students who were actively seeking help when they had no other choices:


*“Students aren’t always quick to ask for help, in my opinion. I suppose they only do that when they really can’t cope anymore.”*
(p3)

Like the respondents in the survey, the interviewees varied in their views regarding their own roles in supporting students with chronically ill family members. All of the participants said that they would help their students to continue their studies, as well as seeking to prevent mental health problems. However, some of the lecturers wondered whether it was their responsibility and were cautious about stepping into a caregiver role:


*“I think you have to be careful not to take on too much of a caregiver role. I try to be mindful of that. I’m trying to help them so they can keep on pursuing their studies, I’m not there to offer support in every area of their lives.”*
(p2)

Most interviewees indicated that supporting these students was not part of their formal role description. Nevertheless, most of them did provide support to these students. They categorized this support as active or passive. All of the interview participants agreed that active support began with listening to the students’ stories:


*“Listening is obviously really important. You need to create opportunities to meet up so you can really listen to the student’s story. That takes some effort, and you obviously have to be able to read between the lines and figure out what’s really going on.”*
(p9)

Most of the participants also mentioned the importance of asking about the wellbeing of the student, rather than simply asking about their chronically ill family member:


*“Sometimes it’s really helpful to just ask the student how they’re doing. That means a lot to them.”*
(p4)

Some participants stated they also wanted to help their students to reflect on their family situations and make choices for their own wellbeing:


*“You want them to reflect on their situation. It’s obviously up to them, though, and I realize it must be quite difficult to deal with that stuff in some cases.”*
(p8)

Other participants chose a more passive supporting approach and made referrals for professional help (e.g., to a student psychologist or general practitioner). All of the participants mentioned that this option was valuable when the problems within the family appeared particularly complex. The participants from the non-health-related educational programs were especially resolute about not providing active support themselves, preferring to refer quickly:


*“I’m not a psychologist or a behavioral scientist, so I realize I have limited expertise. So, if someone has all kinds of complicated problems, I’m not going to try and fix everything myself or give advice.”*
(p12)

As seen among the survey respondents, many participants from health-related educational programs reported providing active support themselves, both because they felt adequately competent to provide active care and because they felt that they were more approachable for students than external professionals might be:


*“Students like the one I talked to this afternoon don’t mind opening up to me, but I’ve been seeing them for six months now, even though it’s all online. They have some idea of who they’re talking to and what they can expect. I think that’s a lot more accessible than getting help from some abstract person who you don’t know.”*
(p1)

All interviewees emphasized the importance of staying in contact with students after referral to other professionals:


*“Sometimes you just need to send an email to ask how things are going, how their doctor’s appointment went or how their sister is doing. Things like that. You just need to show some interest.”*
(p9)

Finally, all of the interviewees mentioned passively supporting their students by being lenient with deadlines, thus allowing them room to continue their studies:


*“You also need to be flexible sometimes and let them hand in assignments some other time or choose a different subject. Things like that.”*
(p13)

In the interviews, the lecturers indicated certain barriers in identifying and supporting students. First, they mentioned the potential “invisibility” of the problem:


*“I think 90% of my colleagues would always be willing to help and throw someone a lifebelt, but they do need to be aware that person is drowning.”*
(p12)

Most of the interviewed lecturers mentioned a two-way battle between their role view and their compassion for their students. In this context, many chose to initiate conversations with their students, despite having limited time to talk:


*“Time is a factor though. I mean, I’m responsible for a lot of students and I only get a limited number of hours to help them. If I did the maths, I’m not sure it would be enough to have a real conversation with everyone.”*
(p5)

This was mentioned in particular as a dilemma for lecturers in the health-care-related educational programs who had identified many of their students as growing up with chronically ill family members:


*“We’ve got all these new students, and I was supposed to have introductory meetings with each of them. You get about half an hour for every meeting. I have to admit, after about 10 students, I started to wonder: is there anyone here who has a normal home life?”*
(p4)

The invisibility of the support options available within the educational institutions was also mentioned. This made making referrals more difficult:


*“Where do you refer students? Who can help them most effectively? What’s the most effective channel? And how can students access support services when they need them?”*
(p11)

All responding lecturers agreed that students with chronically ill family members needed specific support in order to be able to successfully complete their studies. Most respondents agreed or strongly agreed (70.2%, *n* = 146) that supporting students with chronically ill family members was the responsibility of the educational institution, while half (49%, *n* = 102) felt that it was the responsibility of the lecturer.

Similar to most of the survey participants, the interviewees were unclear about the responsibilities that were entailed in their role as lecturer. They all mentioned that the educational institutions had a responsibility to ensure students’ wellbeing. However, the specific requirements of this were unclear, which left all of them feeling personal responsible for the wellbeing of their students:


*“So, who’s responsible in the end? Is it the GP who is actually treating the student’s sister? Is that doctor also responsible for figuring out the whole informal care system? Are we responsible because we’re the ones who notice these issues? None of these things are clear right now, and I’d like to see them addressed at a national level. After all, we all just want to offer effective help.”*
(p3)

Despite the lack of clarity about their formal roles and responsibilities in this regard, all of the interviewees stated that they felt responsible for identifying and supporting these students. It appeared that this sense of responsibility determined their degree of role fulfillment. Finally, most of the participants working in health-related educational programs mentioned feeling responsible because they felt adequately competent to provide some form of active care. However, they also highlighted difficulties with identifying the boundary between their role as lecturer and that of a health care professional. They were all experienced in coaching conversations and knew about the support possibilities. Some engaged in weekly conversations, which was considered too demanding:


*“Alarm bells started going off. In my own head, I mean. After all, you’re a lecturer, you’re part of the education system, you’re not a counselor. I do have a background in counseling and I’ve become quite comfortable with these kinds of conversations over the years, but that can also be a pitfall.”*
(p2)

### 3.2. Competence

Most of the respondents (66.8%, *n* = 139) indicated that they felt competent to support students with chronically ill family members. However, 45.2% (*n* = 94), most of whom were working in non-health-related educational programs, indicated a desire for more training and expertise in identifying and supporting these students. Specifically, the lecturers cited a need for information about support facilities within and outside the institution and training in communication skills.

The interviewees from health-related educational programs indicated that they felt competent to support students with chronically ill family members. They explained about the importance of interpersonal competences such as an open and empathic attitude, which would lower the threshold for students to approach someone to share their stories:


*“Your attitude should reflect the fact that you care about your students; if you don’t, I think you’re in the wrong line of work.”*
(p10)


*“I think you need to have an open attitude so that students feel that they can come to you for help.”*
(p1)

Being observant was also mentioned as an important competence. This was defined by all participants as more than just observing the students. They defined it as having an instinct for problems, even when there was no apparent cause for concern:


*“You need to be able to tell that something is wrong.”*
(p12)

In addition to citing the need for interpersonal competences, all of the interviewees mentioned the importance of conversational skills. These include the ability to listen and to ask open questions, which they explained was necessary to understand a situation and clarify the need for support:


*“Obviously, you need to listen carefully and be able to ask questions in a way that feels safe for the student.”*
(p9)

Finally, professional reflection was mentioned as an important competence. This was defined by the participants as knowing where one’s expertise ends and when to refer to other professionals:


*“Well, I think it’s important to recognize the limitations of my own expertise. If something is outside my competence, I feel I’m responsible for referring the person on and keeping track of the situation.”*
(p4)

Some participants also stated that one should reflect on when to step out of the superordinate role as lecturer to respect the students’ autonomy:


*“You shouldn’t overestimate your own importance, you know. You shouldn’t think ‘I’m the one to solve this, I’ll just go and...’ God no. Those students have been through so much, they’ve seen more institutions than I have.”*
(p5)

### 3.3. Needs

In line with the survey respondents, the interviewees mentioned a wish for more training and greater expertise in identifying and supporting these students. In particular, they highlighted a need for information about referral opportunities and facilities within and outside the institution, communication-skills courses, and information on how to be supportive of this group of students. The participants wanted more knowledge about the target group, as well as concrete guidelines on conducting coaching conversations and making appropriate referrals:


*“Since we’re talking about knowledge, you definitely need to know something about the subject. You need to know about informal care, young informal caregivers, what it actually means to be in that position, and what sort of support is out there. I think it’s also easier to identify those situations and provide support if you actually know something about the subject.”*
(p3)

Not mentioned in the survey but mentioned by all interviewees was the need for peer-to-peer coaching to discuss difficult student cases and learn from each other:


*“I think we should be more focused on peer-to-peer coaching. You really need the opportunity to meet up with other academic counselors and discuss specific cases.”*
(p1)

The need for clear role descriptions was implicitly mentioned in the survey responses, where there were different opinions about their responsibilities. The interview participants cited a need for greater clarity on the roles and responsibilities of lecturers, as well as the division of tasks between lecturers. Some preferred not to engage in coaching conversations, and task division would allow for more experienced and involved lecturers to take these on, while others could then refer students to their more experienced colleagues:


*“I think it would be better if the lecturers were responsible for identifying problems so that the academic counselors can focus on the rest of the process and figure out what steps we need to take.”*
(p4)

## 4. Discussion

This study revealed that lecturers are unclear about their role in identifying and supporting students with chronically ill family members. Their role description is often presented as “supporting students to continue their studies”. Such a broad definition creates confusion about the degree of responsibility that lecturers are expected to assume.

Most of the survey respondents agreed that supporting students with chronically ill family members was a responsibility of “the educational institution”, while only half agreed that supporting students was the responsibility of the lecturer. However, most of the lecturers who participated in the interviews indicated that they did offer support when they encountered a student struggling because of a chronically ill family member. These interviewees described a two-way battle, owing to a lack of clarity about their role fulfillment on the one hand and their compassion for students on the other. In addition, the results of both the survey and the interviews indicate a discrepancy between the lecturers’ role view and role fulfillment. In practice, the provision of support to students with chronically ill family members seems to depend on the discretion of the individual lecturer.

Furthermore, the lecturers in the survey and interviews described difficulties with identifying those students growing up with chronically ill family members. This finding was in line with those of other studies [25]. These students’ lack of visibility is sometimes attributed to a lack of awareness among lecturers [21,25] or a barrier between students and lecturers that prevents the former from opening up [26,27]. Specifically, students may feel vulnerable when initiating conversations about their family situations or may fear having their family story known by lecturers being considered an excuse for not fulfilling school attendance or assessment requirements [26].

The interviewed lecturers in health-related education seem to feel competent in identifying and supporting students with chronically ill family members when the requests for help are fairly simple. This may be because these lecturers are often also trained as nurses or social workers, and such disciplines have high literacy regarding caregiving and its consequences. This finding is in line with that of Swami [28], who concludes that lecturers from health- and behavioral-science-related programs were more likely than other lecturers to demonstrate high depression literacy and therefore to assist students with mental health issues. Moreover, these more “literate” lecturers are also more likely to be approached by students in need of help [21], indicating that some students may feel more comfortable approaching lecturers who they expect to have a greater understanding of their difficult family situations.

The lecturers also indicated that students with chronically ill family members were more likely to be enrolled in health-related educational programs. This could be explained by the concept of a “care identity” described by Becker and Becker [29]. The authors found that young adult caregivers were more likely to be drawn to care-related careers than other students, which may be attributable to their own extensive caring experience and knowledge about illness.

In the survey and interviews, the lecturers also noted the importance of interpersonal competencies, communication skills, and reflection regarding the autonomy of the students. Multiple previous studies have indicated that lecturers with strong interpersonal competencies and communication skills are more likely to effectively support students with chronically ill family members [14,30,31,32]. In the study by Ali and colleagues, the students mentioned a need for immediate and accessible support in times of crisis [31]. In addition, they emphasized the importance of being listened to and encouraged when sharing details of their situations and of being supported to cope with their family situations [14,31,32]. Finally, flexibility regarding deadlines for papers and other educational activities was mentioned in our study, as well as in the study of Kettell [14], described as a passive method of supporting students with chronically ill family members [14].

### 4.1. Strengths and Limitations

Our mixed-methods approach was valuable for gaining rich data regarding the competencies and role views of lecturers. The interview responses highlighted a dilemma: the lecturers felt responsible for their students’ wellbeing but were unclear about their role fulfillment. This dilemma was expressed through strikingly open and sometimes emotional stories shared during the interviews, with the participants stating that they were glad about the attention being given to this topic.

However, given that these survey respondents and interview participants volunteered to participate, it is likely that they had a particular interest—or experiences—in supporting students with chronically ill family members, which may have led to selection bias. In other words, these results might paint a distorted image of lecturers being especially interested and literate in the need to support students with chronically ill family members.

A further limitation of the study is that we only recruited lecturers from universities of applied sciences. Thus, it remains unclear whether lecturers in schools for secondary vocational education would have the same experiences and attitudes. Students in secondary vocational educational programs tend to experience more family problems than those in applied science universities [33]. Therefore, lecturers in vocational education programs may be more familiar with the target group and their needs.

### 4.2. Clinical and Research Implications

In the educational institutions represented by the lecturers in this study, there were no specific professionals tasked with identifying and supporting students with chronically ill family members. Previous studies have asserted that educational institutions should play a major role in identifying affected students [14,32,34] and referring them to external professionals when their support needs are too complex to be handled by school professionals.

Our findings suggest that agreements between education and health care institutions regarding responsibilities for coordination and referrals are necessary to support such students. Lecturers’ formal role descriptions need to be clarified. This would allow support and referral tasks to be better allocated to those lecturers who wish to take on supporting roles for students who have mental health problems—or who are at risk of developing such problems—due to the pressure of caring for chronically ill family members.

Furthermore, the lecturers cited a wish for more peer-to-peer coaching to discuss complex student cases, more training in communication skills, and more information regarding how to identify and support students with chronically ill family members. Therefore, further research regarding the implementation and effects of such training at the beginning of a lecturer’s tenure, with follow-ups in the form of group peer-to-peer coaching, is recommended to further assist lecturers in their supporting role. Research on lecturers in other types of educational institutions is also recommended to assess whether these results are equally applicable elsewhere.

## 5. Conclusions

Lecturers who are aware of the challenges faced by students growing up with chronically ill family members are more inclined to identify and support such students, working to prevent the development of mental health problems and striving to keep the student in education. This research underlines the need for a clear acknowledgment of who is responsible for supporting such students and what support options are available. Furthermore, research on the effects of training at the start of the teaching profession and of follow-up peer-to-peer coaching is required to support lecturers in their role as referrers. In addition, further research is required to determine whether the present results also apply to lecturers without such literacy and those working for other types of educational institutions.

## Figures and Tables

**Figure 1 ijerph-20-04978-f001:**
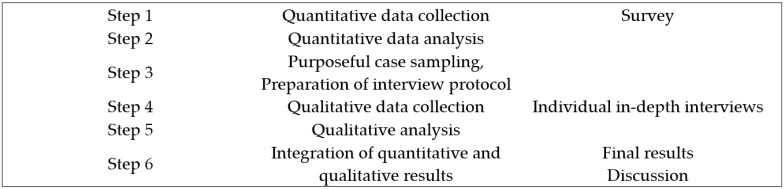
Explanatory sequential design by Cresswell and Plano Clark [22].

**Table 1 ijerph-20-04978-t001:** Background characteristics of the lecturers participating in the survey (N = 208).

Characteristics	N	%
Age (y)		
20–40	81	38.9
>40	127	61.1
Age mean y (SD)	45.2 (11.1)	
Gender		
Female	158	76
Years of teaching experience (y)		
0–5	71	34.1
5–10	48	23.1
>10	89	42.8
School		
Nursing and Social Care Schools	140	67.3
Economics, Sports, Law, Media, Communication and ICT, Arts and Technical Schools	68	32.7

**Table 2 ijerph-20-04978-t002:** Background characteristics of the interviewed lecturers.

Lecturer	Sex	Age (y)	School	Work Experience as a Lecturer (y)
1	Woman	30–40	Social work	5–10
2	Man	50–60	Nursing	10–15
3	Woman	30–40	Nursing	5–10
4	Woman	<30	Nursing	<5
5	Man	40–50	Applied Psychology	10–15
6	Woman	30–40	Social work	<5
7	Man	50–60	Economics	10–15
8	Woman	40–50	Social Work	5–10
9	Man	50–60	Nursing	<5
10	Man	50–60	Nursing	20–25
11	Woman	>60	Sports	25>
12	Man	50–60	Media, Communication and ICT	5–10
13	Woman	30–40	Sports	5–10

## Data Availability

Data on which the conclusions of the manuscript rely are presented in the main paper. Data collected and analyzed during the current study are available from the corresponding author upon reasonable request.

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
