# Peer review of "Identifying and Supporting Students with a Chronically Ill Family Member: A Mixed-Methods Study on the Perceived Competences and Role Views of Lecturers"

_ijerph, 2023, doi:10.3390/ijerph20064978_

Round 1

Reviewer 1 Report

page one: “ conducted by the OrganizationWorld Health Organisation” separate organization and world.

The article is well written and presented, shading light on a crucial problem. Here's my comment:

1) It is not fully clear why the sample of the interview could be compared with the sample of the survey. Descriptive statistics were reported in different ways in the two groups and it seems that the two groups have a different composition under various variables. It could be useful to comment on the differences existing between the group in the survey and the group in the interview.

2) page 2 and 3: please ameliorate the quality of figure one

3) tables: they must be reported by Following apa guidelines (please see here https://apastyle.apa.org/style-grammar-guidelines/tables-figures/tables)

Author Response

Dear reviewer, thank you for your time and effort to review our paper.

Thank you for pointing out the grammar mistakes (1), the quality of the figure (3) and style of the tables (4). We checked it to make sure it is correctly displayed.

Thank you for pointing out the unclarity in the results section (2). We did not compare the sample of the interviewees with the sample of the survey. Qualitative data were collected to obtain a more detailed explanation or underpinning of the quantitative results. We re-arranged the whole result section for more clarity.

Reviewer 2 Report

This is a significant and timely study. However, a few improvements can be made to the paper:
1. The introduction should include more discussion on role views and role fulfillment from previous studies. Previous studies on lecturers' perceived competence and needs in supporting students with chronically ill family members should also be discussed.
2. As this is an explanatory sequential design, it is important to highlight which part of the quantitative findings needs further explanation and how that is achieved through the qualitative approach.
3. The themes in the presentation of the qualitative findings should be re-arranged to make it easier to follow and understand. For example, role view and role fulfillment should be discussed together and tied up with the quantitative findings. Identifying roles, supporting roles, and barriers (in identifying and supporting students) should not be three separate themes but discussed as sub-themes, perhaps under role views. The qualitative findings on competence and needs should support the quantitative findings.
4. There are parts where the formatting seems to be off.
5. The discussion mentioned that lecturers in health-related education to be more competent, but the survey did not indicate this clearly, except that "45% (n = 94), most of whom were working in non-health-related educational programmes, indicated a desire for more training and expertise in identifying and supporting these students". 

Author Response

Dear reviewer, thank you for your time and helpful suggestions to improve our paper.

To the best of our knowledge there are few studies describing the perspectives of professionals around children or young adults with chronically ill family members. These studies are usually oriented to a broad group of professionals such as healthcare professionals or school social workers. However, we have added a recent study that examines teachers teachers’ perspectives on the provision of support for young carers (age < 18 yrs).

We added the following paragraph into the introduction section (1).

A recent study on teachers’ perspectives on the provision of support for young carers (age < 18 yrs) shows the difficulties teachers experiencing while identifying young carers and balancing the provision of extra support within the constraints of the school context (Warhurst et al., 2022).

Thank you for your helpful suggestion to make the results section easier to understand and your comment to highlight which part of the quantitative findings needs further explanation and how that is achieved through the qualitative approach (2&3). We agree that this could be more clear to make it easier to follow. We re-arranged the whole results section to support the quantitative findings by the qualitative data.

Thank you for pointing out the formatting (4). We checked it to make sure it is correctly displayed.

We agree with your comment that this statement is not fully indicated from the survey (5). We followed your suggestion and have rephrased the discussion with: ‘The interviewed lecturers in health-related education seem to feel competent in identifying and supporting students with chronically ill family members when the requests for help are fairly simple.’
